# Targeting the Pulmonary Microbiota to Fight against Respiratory Diseases

**DOI:** 10.3390/cells11050916

**Published:** 2022-03-07

**Authors:** Zongjie Li, Yuhao Li, Qing Sun, Jianchao Wei, Beibei Li, Yafeng Qiu, Ke Liu, Donghua Shao, Zhiyong Ma

**Affiliations:** Shanghai Veterinary Research Institute, Chinese Academy of Agricultural Science, Shanghai 200241, China; lizongjie@shvri.ac.cn (Z.L.); liyuhaoaiwojia@163.com (Y.L.); sun7355608@outlook.com (Q.S.); jianchaowei@shvri.ac.cn (J.W.); lbb@shvri.ac.cn (B.L.); yafengq@shvri.ac.cn (Y.Q.); liuke@shvri.ac.cn (K.L.); shaodonghua@shvri.ac.cn (D.S.)

**Keywords:** lung microbiota, gut-lung axis, immunity homeostasis, inflammatory response, respiratory disease

## Abstract

The mucosal immune system of the respiratory tract possesses an effective “defense barrier” against the invading pathogenic microorganisms; therefore, the lungs of healthy organisms are considered to be sterile for a long time according to the strong pathogens-eliminating ability. The emergence of next-generation sequencing technology has accelerated the studies about the microbial communities and immune regulating functions of lung microbiota during the past two decades. The acquisition and maturation of respiratory microbiota during childhood are mainly determined by the birth mode, diet structure, environmental exposure and antibiotic usage. However, the formation and development of lung microbiota in early life might affect the occurrence of respiratory diseases throughout the whole life cycle. The interplay and crosstalk between the gut and lung can be realized by the direct exchange of microbial species through the lymph circulation, moreover, the bioactive metabolites produced by the gut microbiota and lung microbiota can be changed via blood circulation. Complicated interactions among the lung microbiota, the respiratory viruses, and the host immune system can regulate the immune homeostasis and affect the inflammatory response in the lung. Probiotics, prebiotics, functional foods and fecal microbiota transplantation can all be used to maintain the microbial homeostasis of intestinal microbiota and lung microbiota. Therefore, various kinds of interventions on manipulating the symbiotic microbiota might be explored as novel effective strategies to prevent and control respiratory diseases.

## 1. Introduction

Over the past two decades, accumulated studies on the interactions between lung microbiota and the host immune system have provided invaluable understanding about the immune modulating functions of the lung microbiome [1,2]. The lungs of healthy individuals have been considered to be sterile for a long time according to the classic respirology theory. However, the development of culture-independent sequencing technology has proved that there are abundant and diverse microbial communities in the respiratory tract and have intimate associations with the host’s health and disease [3,4,5]. The acquisition and maturation of the lung microbiome in the early life can be influenced by delivery mode, feeding practices, living environment, and other affecting factors [6]. The interactions among the lung microbiota, infected viruses, invading bacteria, and the host immune system can affect the susceptibility to lower respiratory tract infections and diseases [7]. Therefore, investigations on the microbial composition and the immune function of the lung microbiome may provide novel effective interventions and preventive measures to treat respiratory diseases.

The respiratory tract has a large exposed surface to execute the air-exchange function, while the abundance and diversity of the lung microbiome are obviously influenced by the contacting air environment [8]. Though the lungs of mammals are equipped with an effective antimicrobial defense system, the relatively lower microbial biomass of the pulmonary microbiota successfully colonized the respiratory tract and obtained tolerance to the host immune system [9,10]. Moreover, the stably harbored lung commensal bacteria can generate colonizing resistance and help to fight against the invading outer pathogens by producing various kinds of antimicrobial molecules. Therefore, the lung microbial ecosystem is maintained by the balance of transiently entered and selectively eliminated microorganisms [11]. The bi-directional cross-talk between the lung microbiome and the host immune system plays a fundamental role in keeping the lung immune homeostasis. On one side, the inhabited pulmonary microbiota can influence the maturation of the host immune system by producing numerous structural ligands and metabolites (such as lipopolysaccharide, peptidoglycan, and short-chain fatty acids). On the other side, the host’s innate and adaptive immune system can alter the lung microbiome by forming biophysical barriers, secreting immunoglobulin A (IgA), producing antimicrobial peptides, and recognizing the resident and viable microbes [12]. Therefore, many factors that cause lung microbiome dysbiosis can alter the pulmonary immune homeostasis and induce the occurrence and development of respiratory inflammation and diseases [13].

Increasing evidence has revealed that the disturbed balances of intestinal microbiota and lung microbiota had intimate correlations and can corporately cause respiratory diseases. The comprehensive and sophisticated interactions between intestinal microbiota and lung microbiota and their collective actions in modulating the pulmonary immune homeostasis might provide new therapeutic targets and manipulating strategies for clinical treatment of respiratory infections [14,15]. Research about the gut-lung axis demonstrated the depletion of the gut microbiota in C57BL/6 mice could induce lung bacterial dissemination, organ damage and enhance mice mortality during *Streptococcus pneumoniae* infections. However, the restored process of the gut microbiota by fecal microbiota transplantation (FMT) could reverse the survival rate of broad-spectrum antibiotics treated mice by regulating alveolar macrophage function and inflammation response [16,17]. Probiotics treatments targeting gut and lung microbiota could confer health benefits for the host during the chronic lung disease progression, and other interventions to protect the microbial ecosystem balance of the gastrointestinal tract and the respiratory tract could also be exploited to protect the respiratory immune system [18,19,20].

In this review, we mainly summarize the composition of respiratory microbiota and their potential functions related to the host immune system. Moreover, we also discuss the novel therapeutic approaches targeting the gut and lung microbiota to treat respiratory diseases.

## 2. The Origin of Pulmonary Microbiota and the Influencing Factors

The lungs of healthy individuals have previously been considered to be sterile because the traditional microbiology approaches usually cannot give out positive cultural results [21]. Progress on the culture-independent techniques has proved that the colonization of microbial community in respiratory tract began immediately after birth, for the reason that the bacterial DNA in the tracheal aspirates of healthy neonates could be detected 24 h after birth [22]. By comparing the microbial communities in oral wash, nasal swab, and bronchoalveolar lavage (BAL) from the healthy subjects, the bacterial communities of the lungs shared much higher similarity with those from the oral cavity, but were different from the nasal cavity. Therefore, the lung microbiota might quite possibly originate from the oral microbiota through the microaspiration, which usually occurred during the sleep process when the tone of the oral and pharyngeal muscles is diminished [23].

Initial microbial colonization in the respiratory tract is mainly impacted by delivery mode, antibiotic usage, dietary structure, environment exposure, and pathogenic infections (Figure 1) [24,25,26,27]. In early life, caesarean delivery patterns, increasing use of antibiotics, changes in food composition, and the contacting environmental microorganisms can all directly and indirectly impact the diversity and abundance of the lung microbiome [28,29,30]. Moreover, pathogenic infections induced by various kinds of viruses can also play important roles in shaping lung microbiota formation [31,32]. Comprehensive analysis of the environmental and lifestyle factors that influence the early colonization of the lung microbiome might provide preventative interventions and therapeutic strategies to treat respiratory diseases [33,34].

## 3. The Diversity and Composition of Respiratory Microbiota

At birth, the acquisitions of infant microbial community in the mucosal surfaces is mainly determined by the microbes derived from the mother’s vagina, skin and intestinal tract [35,36]. In the subsequent early life period, the distribution, composition and development of respiratory microbiota are transiently diversified together with the maturation of the host immune system [37,38]. As the air-exchange area, the lung environmental condition is vastly different from other body sites, therefore the compositions and diversities of lung microbiota are mainly determined by the transient change of entering outer microorganisms and selective elimination of viable microorganisms. During the microbe–host crosstalk process, the host respiratory tract has developed a variety of selective strategies to maintain the balance of the microbial ecosystem [39].

Though the lung has a large surface to directly contact with the outer air environments, the pulmonary immune system is equipped with an effective antimicrobial and defensive system to fight against the invading foreign microorganisms. Therefore, the microbial biomass in the lung is remarkably lower than that in other body sites [7,40]. At the phylum level, the predominant microbial communities in the lungs are mainly composed of Proteobacteria, Firmicutes, Tenericutes, and Bacteroidetes. When analyzed at the genera level, the most common genera in the lungs of healthy individuals are mainly composed of *Prevotella*, *Veillonella*, *Streptococcus* and *Pseudomonas*. When compared with the adjacent sites, enhanced richness of *Proteobacteria*, *Ralstonia* and *Haemophilus* and decreased abundance of *Prevotella*-affiliated taxa are consistently observed, the unique compositions of lung microbiota are possibly related to the redox state and oxygen application of the lower respiratory tract [41,42,43]. The composition of lung microbiota in health and disease states differed apparently when the balance of host immune response is disturbed by viruses, allergens or genetic deficiency. Upon the conditions of cystic fibrosis (CF), chronic obstructive pulmonary disease (COPD), and other chronic lung diseases, the predominant genera were shifted to *Pseudomonas*, *Streptococcus*, *Prevotella* and *Haemophilus*. When compared with the healthy individuals, the richness of *Bacteroidetes* in the patients was significantly decreased [7]. Studies about the lung microbiota in patients with asthma indicated that the abundance of *Proteobacteriae* was increased which might be driven by the *Haemophilus*, *Moraxella* and *Neisseria* species [44]. Multiple studies demonstrated that gut dysbiosis was observed in patients with coronavirus disease 2019 (COVID-19), and the gut microbiota richness and composition of COVID-19 patients were quite different from those of healthy controls. After severe acute respiratory syndrome coronavirus 2 (SARS-CoV-2) infection, the richness and diversity of gut microbiota were both significantly decreased, and microbial richness could not restore to normal levels even after 6-month recovery. However, the predominant microbial taxa of severely ill patients were characterized by *Burkholderia cepacia* complex, *Staphylococcus epidermidis*, or *Mycoplasma spp.* (including *M. hominis* and *M. orale*) [45,46,47]. 

## 4. Relations between the Gut Microbiota and Lung Microbiota Mediated by the Gut-Lung Axis

The “Zang-Fu” theory in traditional Chinese medicine describes that “lung and large intestineare interior-exteriorly related”, which demonstrates the close physiological and pathological connections between the gut and lung [48]. In fact, the microbiota–host communications can transmit multiple intestinal signals to different distal organs and contribute to host health and disease [49,50]. By comparing the microbial community structures between lung and intestine bacteria, evidence reveals that members of lung and intestine bacteria can directly exchange through the lymph circulation [51,52,53]. Additionally, gut microbiota can produce various kinds of bioactive metabolites (such as butyrate, p-cresol sulfate, and indoles) to impact the host immune response and energy homeostasis [54,55,56,57,58]. The innate lymphoid cells (ILCs) derived from intestinal lamina propria have important action on host defense and inflammatory responses. When the interleukin-25-induced group 2 innate lymphoid cells (ILC2) and interleukin-22 (IL-22)-producing group 3 innate lymphoid cells (ILC3) migrate to the lung, the host resistance to pneumonia and other inflammatory infections could be promoted [59,60,61,62]. Therefore, the gut microbiome plays an important role in regulating pulmonary immune function and health protection through signals transmitted by the gut–lung axis.

The regulating role of the lung microbiota on intestinal infectious diseases through the gut–lung axis should also not be neglected (Figure 2). Alteration of pulmonary microbiota is found to be able to modulate the microbial communities of gut and influence intestinal immunity and disorders [63]. Pulmonary infections caused by *Mycobacterium tuberculosis* can induce a distinct dysbiosis of the gut microbiota by decreasing the α-diversity; however, the anti-tuberculosis therapy can also cause a rapid and significant change in the diversity and composition of the gut microbiota [64,65,66,67,68,69]. Recently, the gut microbiota is considered to be a novel potential therapeutic target in treating tuberculosis. Supplementation of *Lactobacillus* could restore the dysregulated the gut microbiota and enhance the lung dendritic cells (DCs) function and subsequent T cell response to control tuberculosis [70]. Moreover, the diversity of the intestinal microbial community can be significantly disturbed in the *Pneumocystis murina* and *Klebsiella pneumoniae* caused respiratory infection [71,72]. The dysbiosis of gut microbiota and subsequent dysregulation of microbiota-related immunological processes could also be observed in patients with asthma, COPD, and other chronic respiratory diseases [73,74]. Therefore, precise treating approaches to modify the lung and gut microbiome might play important roles in the management and treatment of respiratory diseases.

## 5. The Lung Immune Homeostasis Shaped by Microbe-Host Interactions

The crosstalk between the pulmonary microbiota and host mucosal immune system plays a fundamental role in maintaining lung immune homeostasis [12]. Numerous factors that cause pulmonary microbiota dysbiosis could disturb the immune function and induce inflammation responses and airway diseases [34]. The “hygiene hypothesis” explained the critical relevance between the symbiotic microbiota alteration and the immune homeostasis dysregulation, because the modern lifestyle in industrialized societies altered the human microbial ecosystem and increased the occurring chances of infectious disease [75,76]. In early life, the formation and development of the commensal microbiota have critical impacts on the maturation of the immune system.

The microorganism-associated molecular patterns (such as lipopolysaccharide and flagellin) of commensal microbes can induce the production of secretory immunoglobulin A (sIgA) and establish the balance of immune recognition and immune tolerance [77]. During the process of pregnancy, the pattern of the immune system in the fetal environment is dominated by the Th2 phenotype. When the neonate’s symbiotic microbiome is acquired after birth, the polarization of lung naïve T cells begins to shift from Th2 to Th1 phenotype, and then the infant`s resistance to allergic diseases is enhanced [78,79,80]. Because the microbial compounds could induce the differentiation of regulatory T cells (Treg) and Th17 cells, the dysregulated lung microbiota by bleomycin treatment induce the production of interleukin-17B (IL-17B) and tumor necrosis factor-α (TNF-α) through Toll-like receptor-Myd88 adaptor signaling [81,82]. Therefore, manipulation of the infant microbial community could train the responses of the innate and adaptive immune system, and provide promising approaches to benefit life-long health [83,84].

## 6. The Lung Microbiota: Potential Target to Prevent and Treat Pulmonary Infections

The microbial communities harbored in the lung can build a protective barrier against respiratory diseases, while the complicated interactions among the pulmonary microbiome, pathogenic virome, and host immunity act as critical roles in lung inflammation and immune responses [85,86]. The secondary bacterial infections often happen together with or after viral infections, and the infectious mechanism can be explained the dysregulated innate and acquired immune homeostasis and the pathological damage which are caused by the airway tract viruses. Conversely, persistent infection or colonization of pathogenic bacteria can also induce viral infections by increasing the expression of viral entry receptors [87,88,89]. Emerging evidence has proved that the sophisticated interactions between viruses and bacteria at multiple levels in the lower respiratory tract can influence the host phenotypic effects; therefore, investigations on the viral and bacterial co-infections help to explore novel effective approaches for preventing respiratory diseases [90,91,92].

Recent researches on the lung microbiome promote the understanding of the therapeutic target of commensal microbiota for various kinds of respiratory diseases. Numerous probiotics have been widely applied to treat infectious airway diseases for the beneficial role of regulating the host immunity and inhibiting the invasion of the pathogen [93]. Accumulated studies have demonstrated that oral or nasal administration of *Lactobacillus* and other probiotics could modulate the respiratory innate immune responses and promote health benefits against influenza virus, and several other lactic acid bacteria (LAB) strains were also reported to be able to stimulate the mucosal immune system and provide effective protection against *Streptococcus pneumoniae* infections [94,95,96,97]. The symbiotic microbiota can systematically impact the host respiratory system against *Klebsiella pneumoniae* infection and produce short-chain fatty acids (SCFAs) and subsequently activate the G protein-coupled receptors (GPCRs). The metabolic SCFAs could protect against syncytial virus (RSV) infection by involving and engagement of interferon-β (IFN-β) via the IFN-1 receptor (IFNAR) signaling [98,99,100]. The enhanced anti-viral function of alveolar macrophage was mainly derived by the up-regulation of IFN-β, and then the recovery of the lung pathology was also promoted [101,102]. According to the fact that the gut microbiota dysbiosis was involved in the magnitude of COVID-19 severity through modulating the host immune responses, targeted manipulation to restore the gut microbiota could be an important strategy to treat COVID-19 and speed up recovery [103,104,105]. Nutritional intervention may play a prominent role in establishing and regulating the compositions of intestinal and lung microbiome, therefore the applications of probiotics, prebiotics and functional foods can prevent or alleviate respiratory infections by directly inhibiting the growth of pathogens or indirectly modulating the host’s immune function (Figure 3) [106,107].

## 7. Conclusions

Recent research on the lung microbiome revealed its immune regulating functions and the protective role in fighting against respiratory diseases. The predominant members of the lung microbiome can comprise a microbial barrier to inhibit the colonization of invading pathogenic microorganisms, and the beneficial metabolites produced by the lung microbiome can enhance respiratory immunity and prevent the occurrence of respiratory diseases. Moreover, the intimate relations between intestinal microbiota and lung microbiota provide new therapeutic targets for clinical treatments of respiratory infections. Probiotics, prebiotics, functional foods and fecal microbiota transplantation can all be applied to maintain the microbial homeostasis of intestinal microbiota and lung microbiota. In all, various kinds of interventions targeting the symbiotic microbiota can be used as novel strategies to prevent and control the respiratory diseases.

## Figures and Tables

**Figure 1 cells-11-00916-f001:**
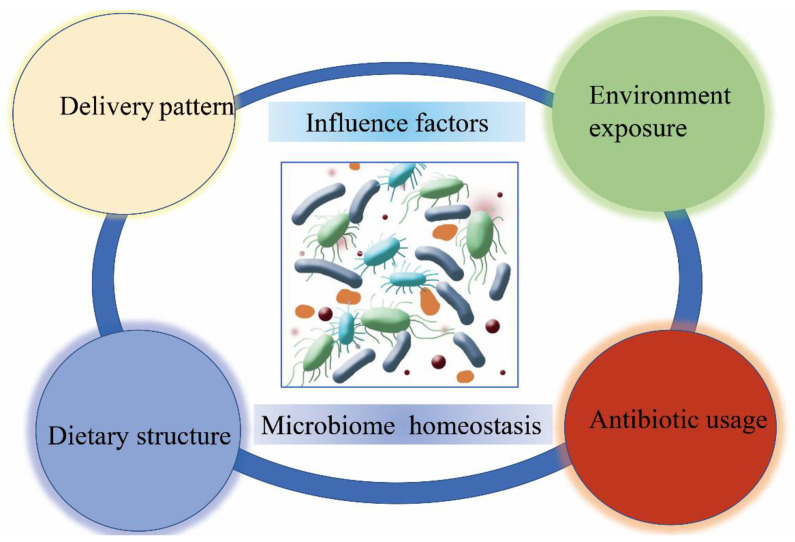
The acquisition and maturation of symbiotic microbiome. The colonization of symbiotic microbiome mainly impacted by delivery mode, antibiotic usage, dietary structure, environment exposure, pathogenic infections, and other affecting factors.

**Figure 2 cells-11-00916-f002:**
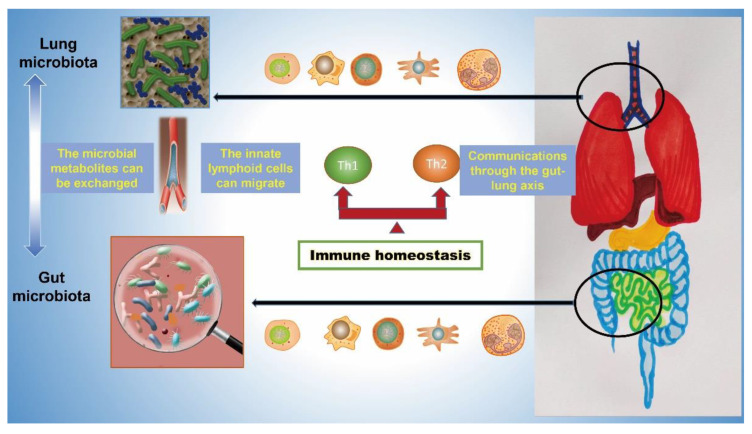
Cross-talk between the gut and lung through the gut-lung axis. Communications between the gut and lung can be realized by multiple signals transmitted from intestine to the lung. Firstly, certain microbial species of lung and intestine microbiota can directly exchange through the lymph circulation. Secondly, various kinds of bioactive metabolites produced by the gut microbiota and lung microbiota can be changed via blood circulation. Additionally, the innate lymphoid cells (ILC2 and3) can migrate from intestinal lamina propria to the lung.

**Figure 3 cells-11-00916-f003:**
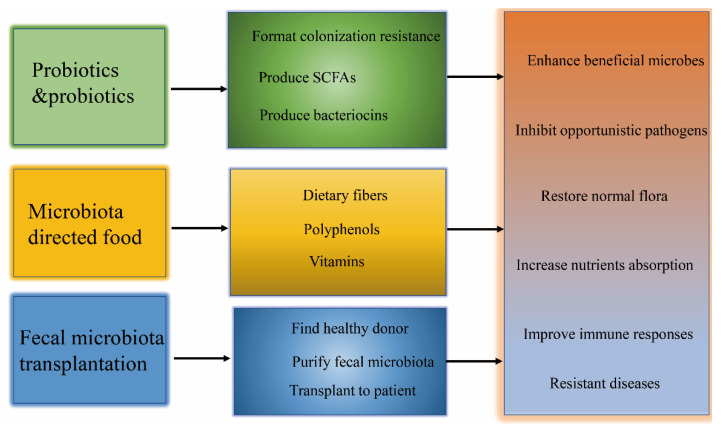
Strategies targeting the symbiotic microbiota to prevent and control respiratory diseases. Probiotics consumption, nutrient intervention, and fecal microbiota transplantation targeting the gut and lung microbiota could confer health benefits for respiratory diseases.

## Data Availability

Not applicable.

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
