# Peer review of "Targeting the Pulmonary Microbiota to Fight against Respiratory Diseases"

_cells, 2022, doi:10.3390/cells11050916_

Round 1

Reviewer 1 Report

Dear Authors,

Thank you for your submission.

The paper is excellent and exciting. I have one comment. While reading the article, I did not think your conclusion would have concentrated on the COVID-19 pandemic. I am not sure if left as it is; the conclusion section releases the right message. I understand that this may be a form of treatment for COVID-19, but I believe you should put that in context. Perhaps you should review the conclusion section.

Author Response

Dear Reviewer,

Too much thanks for the expert comments concerning our manuscript. Those comments are all valuable and very helpful for revising and improving our paper. We have studied comments carefully and have made corrections which we hope meet with approval. The main corrections in the paper and responds to the comments are as following:

Point 1: The paper is excellent and exciting. I have one comment. While reading the article, I did not think your conclusion would have concentrated on the COVID-19 pandemic. I am not sure if left as it is; the conclusion section releases the right message. I understand that this may be a form of treatment for COVID-19, but I believe you should put that in context. Perhaps you should review the conclusion section.

Response 1: The the conclusion part has been changed as required, and treatment methods for COVID-19 had been added in the context.

Reviewer 2 Report

-Li et al., have chosen an important topic; however, most parts of the manuscript are generalized and repetitious. Authors must also have their manuscript proofread by someone experienced in academic writing. 

-Start from the abstract : 
-The abstract section does not give proper information. Abstract means a full-fledged summary that should highlight the information and topics covered in the manuscript. Please revise the abstract (needs to rephrase and rewrite some sentences). Also, highlight essentialities
and future perspectives of the study.

-Line 7 > an effective 
-Line 7-9> either break it into two sentences or write it as >... "The mucosal immune system of respiratory tract possesses effective “defense barrier” against the invading pathogenic microorganisms; therefore, the lung of a healthy organism is considered to be sterile for a long time according to the strong pathogens-eliminating ability."

-Line 9:  The emergence ... 

-Line 13: ...environmental exposure and antibiotic usage. However, the formation....

-Line 18: control respiratory diseases..

-Likewise, the rest of the manuscript must be proofread by an expert. 

-Authors should discuss the microbiome of lungs at normal and diseased state. 

-Figures must be improved in quality. 

-The conclusion section needs to be more elaborative and should highlight the importance of the study and future directions with possible limitations.

Author Response

Dear Reviewer,

Too much thanks for the expert comments concerning our manuscript. Those comments are all valuable and very helpful for revising and improving our paper. We have studied comments carefully and have made corrections which we hope meet with approval. The main corrections in the paper and responds to the comments are as following:

Point 1: Li et al., have chosen an important topic; however, most parts of the manuscript are generalized and repetitious. Authors must also have their manuscript proofread by someone experienced in academic writing. 

Response 1: The academic writing has been carefully improved.

Point 2: The abstract section does not give proper information. Abstract means a full-fledged summary that should highlight the information and topics covered in the manuscript. Please revise the abstract (needs to rephrase and rewrite some sentences). Also, highlight essentialities and future perspectives of the study.

Response 2: The abstract has been corrected to summarize the core information and main topic.

Point 3: -Line 7 > an effective

-Line 7-9> either break it into two sentences or write it as >... "The mucosal immune system of respiratory tract possesses effective “defense barrier” against the invading pathogenic microorganisms; therefore, the lung of a healthy organism is considered to be sterile for a long time according to the strong pathogens-eliminating ability.".

Response 3: This part has been corrected as required.

Point 4: -Line 9:  The emergence ...

Response 4: This part has been corrected as required.

Point 5: -Line 13: ...environmental exposure and antibiotic usage. However, the formation....

Response 5: This part has been corrected as required.

Point 6: -Authors should discuss the microbiome of lungs at normal and diseased state.

Response 6: The comparisons of microbiome in healthy subjects and patients with COVID-19 were added.

Point 7: -Figures must be improved in quality.

Response 7: All the figure qualities have been improved.

Point 8: -The conclusion section needs to be more elaborative and should highlight the importance of the study and future directions with possible limitations.

Response 8: The conclusion section has been rewritten

Round 2

Reviewer 2 Report

I'm still not satisfied with the figure quality. I will leave that to handing editor.